# The ALS-Associated FUS (P525L) Variant Does Not Directly Interfere with Microtubule-Dependent Kinesin-1 Motility

**DOI:** 10.3390/ijms22052422

**Published:** 2021-02-28

**Authors:** Anne Seifert, Hauke Drechsler, Julia Japtok, Till Korten, Stefan Diez, Andreas Hermann

**Affiliations:** 1Department of Neurology, Technische Universität Dresden, 01307 Dresden, Germany; Anne.Seifert@uniklinikum-dresden.de (A.S.); Julia.japtok@uniklinikum-dresden.de (J.J.); 2German Center for Neurodegenerative Diseases (DZNE), 01307 Dresden, Germany; 3B CUBE—Center for Molecular Bioengineering and Cluster of Excellence Physics of Life, Technische Universität Dresden, 01307 Dresden, Germany; hauke.drechsler@tu-dresden.de (H.D.); till.korten@tu-dresden.de (T.K.); 4Max Planck Institute of Molecular Cell Biology and Genetics, 01307 Dresden, Germany; 5Translational Neurodegeneration Section “Albrecht-Kossel”, Department of Neurology, University Medical Center, University of Rostock, 18147 Rostock, Germany; 6German Center for Neurodegenerative Diseases (DZNE), Rostock/Greifswald, 18147 Rostock, Germany; 7Center for Transdisciplinary Neurosciences Rostock (CTNR), University Medical Center, University of Rostock, 18147 Rostock, Germany

**Keywords:** motor neurons, axonopathy, molecular motors, axonal transport, gliding motility assays

## Abstract

Deficient intracellular transport is a common pathological hallmark of many neurodegenerative diseases, including amyotrophic lateral sclerosis (ALS). Mutations in the fused-in-sarcoma (FUS) gene are one of the most common genetic causes for familial ALS. Motor neurons carrying a mutation in the nuclear localization sequence of FUS (P525L) show impaired axonal transport of several organelles, suggesting that mislocalized cytoplasmic FUS might directly interfere with the transport machinery. To test this hypothesis, we studied the effect of FUS on kinesin-1 motility in vitro. Using a modified microtubule gliding motility assay on surfaces coated with kinesin-1 motor proteins, we showed that neither recombinant wildtype and P525L FUS variants nor lysates from isogenic ALS-patient-specific iPSC-derived spinal motor neurons expressing those FUS variants significantly affected gliding velocities. We hence conclude that during ALS pathogenesis the initial negative effect of FUS (P525L) on axonal transport is an indirect nature and requires additional factors or mechanisms.

## 1. Introduction

Intracellular transport of proteins, vesicles or even whole organelles is essential to cell survival. This applies in particular to (motor) neurons, which have to facilitate long-distance transport along their axonal projections with lengths of up to one meter [1,2]. Long-distance transport is driven by molecular motor proteins that translocate along the polarized axonal microtubule cytoskeleton [3]. Anterograde transport (i.e., away from the cell body) along microtubule bundles is driven by plus-end directed kinesin motor proteins, such as the most abundant kinesin-1 [4], while retrograde transport (i.e., towards the cell body) is driven by minus-end directed cytoplasmic dynein. Factors that can differentially affect the motility of these cargo-associated motors include post-translational modifications of the underlying microtubule tracks [5,6,7,8] and the presence of microtubule associated proteins (MAPs) [9,10,11]. One example of the latter is tau, a MAP which is primarily expressed in the central nervous system [12]. Altered tau expression profiles are associated with several neurological diseases like e.g., Alzheimer’s Disease (AD) and Fronto-temporal dementia (FTD), causing axonal transport perturbations early in disease progression [13,14].

Likewise, impaired axonal transport characterizes many other neurodegenerative diseases, including amyotrophic lateral sclerosis (ALS) [15,16]. ALS is the most common motor neuron disease affecting specifically upper (cortical) and lower (spinal) motor neurons. In most cases, it leads to death due to respiratory failure within 2–5 years after symptom onset [17]. About 5–10 % of ALS cases are familial and are caused by specific mutations [18], the most prevalent genes affected are *chromosome 9 open reading frame 72* (C9ORF72, ~40% of familial cases), *superoxide dismutase 1* (SOD1, ~12%), *TAR DNA-binding protein 43* (TDP43, (~4%), or *fused in sarcoma* (FUS, ~5% of familial and 1% of sporadic cases) (reviewed in [19,20]).

FUS is a ubiquitously expressed DNA/RNA binding protein that typically localizes to the nucleus. It is involved in DNA repair, transcription regulation, RNA-splicing/transport [21] and forms liquid droplets with RNA and other proteins (RNP granules) at physiological conditions in the cytoplasm [22]. ALS-associated mutations in FUS have been mapped to almost all its domains [23], with the highest prevalence in the C-terminal nuclear localization sequence (NLS, i.e., R521C and P525L) [20,24]. NLS-FUS variants mislocalize to the cytoplasm [25,26] and cause severe impairment of axonal mitochondrial transport in a *Drosophila* model of ALS as well as in spinal motor neurons that were differentiated from ALS-patient-derived induced pluripotent stem cells (iPSCs) [27,28,29]. Moreover, cytoplasmic FUS associates with stress granules [30,31]—a process aggravated by the mislocalization of FUS variants carrying mutations in the NLS-domain. FUS-NLS variants further cause a liquid-to-solid phase transition of RNP and stress granules, leading to the formation of insoluble aggregates in vivo and in vitro [32,33]. These aggregates sequester, and thereby deplete, RNA and proteins in the cytoplasm, including those of the axonal transport machinery [34]. Further, non-aggregated cytoplasmic FUS has previously been isolated as part of an RNA‑transporting granule associated with kinesin‑1 [35], indicating that non-aggregated FUS may be directly involved in regulating motor motility. In line with this, cytoplasmic granules containing mutant FUS variants colocalize with stress granule markers and kinesin‑1 mRNA and protein [36]. Whether FUS variants directly interact with microtubules or motors has however not been determined to date.

We hence hypothesized that mislocalized pathogenic FUS variants might causally contribute to the observed axonal transport phenotypes by direct interaction of free FUS with the transport machinery (e.g., directly inhibiting motor proteins or acting as road blocks on microtubules). To test this hypothesis, we investigated whether there is any impact of wildtype FUS and FUS-P525L on anterograde axonal transport, exemplified by in vitro reconstituted microtubule gliding motility assays on surfaces coated with kinesin-1 motors. Interestingly, we did not observe any significant changes in microtubule gliding velocities, neither in the presence of recombinant wildtype FUS-GFP or FUS-P525L-GFP nor in the presence of whole‑cell lysates obtained from ALS-patient-specific iPSC-derived spinal motor neurons which endogenously express these FUS variants. We hence conclude that the effect of the ALS-associated FUS-P525L variant on axonal transport is indirect, requiring additional factors or mechanisms that remain to be identified.

## 2. Results

To study the impact of ALS-associated FUS variants on the axonal transport machinery in a minimal system, we adapted a previously established kinesin-1-dependent in vitro microtubule gliding motility assay [37]. In this assay, the gliding of stabilized fluorescently-labeled microtubules on a layer of surface-immobilized kinesin-1 motors is used as readout for collective motor motility in dependency of external factors like ionic strength, temperature, microtubule/motor interacting proteins, and cell lysates.

While microtubule gliding motility studies with recombinant protein were not critical, studies with whole cell lysates of iPSC-derived neurons turned out to be challenging. Whole-cell lysates of iPSC-derived matured motor neurons were prepared by shearing the cells (Figure 1A) in a syringe needle of 400 µm diameter (see materials and methods for details), in order to be applicable in our kinesin 1 dependent microtubule gliding assay. Other lysis methods like dounce homogenization, glass bead shearing, and liquid nitrogen grinding [38,39,40] did not yield sufficient protein for the assay, while buffers commonly used for chemical lysis caused microtubules to detach from the surface-bound motors (Appendix A). Lysates were prepared in PBS supplemented with 10% glycerol (*v/v*), 10 mM β-glycerophosphate and a broad spectrum protease inhibitor mix. Competitive inhibition of phosphatase activity by an excess of β-glycerophosphate is required to limit protein dephosphorylation in the lysates, since other phosphatase inhibitors (e.g., orthovanadate) also inhibit kinesin-1 motor activity and are therefore not compatible with microtubule gliding motility assays (Appendix A and [41]). The presence of β-glycerophosphate in the assay mix, however, caused microtubules to detach from surface-bound motors (Appendix A, white arrow), which is why we added 0.3 % methylcellulose to the final assay mix (Appendix A). Methylcellulose acts as a cushion to keep microtubules close to the surface‑bound motors without affecting their gliding velocity [42,43]. Using epifluorescence microscopy, we observed microtubule gliding in glass flow chambers that were functionalized with surface-immobilized, constitutively-active *Drosophila melanogaster* kinesin-1 (DM‑KHC). Since the microtubule gliding velocity is highly sensitive to changes in temperature [44], flow chambers were fixed onto a Peltier element to minimize temperature changes during the experiment and between single technical replicates. We recorded 10 s time-lapse movies at an acquisition rate of one frame per second and determined the median frame-to-frame velocities of the gliding microtubules (data pooled from ten different fields of view) before and after the addition of recombinant proteins or cell lysates using an automated filament tracking algorithm [45,46] (Figure 1B,C,D). With this setup at hand, we were able to obtain highly reproducible velocity measurements (Appendix A).

### 2.1. The Optimized Microtubule Gliding Motility Assay Detects Recombinant Road-Block Proteins in the Low Nanomolar Range

To estimate the sensitivity of our optimized microtubule gliding motility assay, we titrated recombinant full-length human tau (2N4R tau-GFP) into our final assay mix (Figure 2A). Microtubule-associated tau inhibits kinesin-1 stepping by a simple road-block mechanism [8], significantly reducing the velocity and run-length of kinesin-1 [47]. We supplied recombinant 2N4R tau-GFP at final concentrations of 1 to 500 nM and assessed microtubule-binding qualitatively by epifluorescence widefield microscopy prior to recording 10 s time-lapse movies of microtubule gliding. While no significant 2N4R tau-GFP microtubule binding was evident at low concentrations, we observed binding at 50 nM and above (Figure 2B). The strong background signal originating from free 2N4R tau-GFP in solution, however, prevented us from quantifying the exact amounts of microtubule-bound tau. In line with the observed tau binding, microtubule gliding was not affected below 15 nM 2N4R tau-GFP, but considerably slowed down at 25 nM and completely ceased at concentrations above 50 nM (Figure 2C). We hence conclude that the detection limit of our optimized microtubule gliding motility assay ranges between 15 nM and 50 nM 2N4R tau (equivalent to 1.6–5.3 ng/µL).

### 2.2. Recombinant FUS Does Neither Directly Bind to Microtubules nor Directly Inhibit Kinesin-1 Motility

We next aimed to determine the effect of wildtype FUS-GFP and ALS-associated FUS-P525L-GFP on kinesin-1-dependent microtubule gliding. We first asked whether recombinant FUS has any direct effect on kinesin-1 driven transport by adding increasing amounts (5 nM, 500 nM and 5000 nM) of recombinant GFP-labeled wildtype FUS or FUS-P525L to our kinesin-1-dependent microtubule gliding motility assay (Figure 3A). Interestingly, we did neither observe any significant microtubule-binding (Figure 3B and Appendix A) nor any inhibitory effect on kinesin-1-dependent microtubule gliding (Figure 3C). We hence conclude that neither wildtype FUS nor FUS-P525L directly interferes with kinesin-1-driven transport over a wide range of concentrations including concentrations much above the in vivo situation (Appendix A).

In vivo, cytoplasmic FUS can form aggregates [22] that develop over a time course of hours from liquid compartments in the cytoplasm undergoing a liquid-to-solid phase transition. A similar FUS aggregation was recently shown to also occur at high ionic strength in vitro, but at a much faster time scale (i.e., within seconds, [33]). While we did not observe any significant FUS aggregation at the time scales of our kinesin-1-dependent microtubule gliding motility assays, we cannot exclude that—like in vivo—aggregates which have a direct inhibitory effect on kinesin-1 motility might form over longer time scales. We therefore investigated kinesin-1-dependent microtubule gliding over a time course of three hours. While the overall relative microtubule velocities slightly declined over time, we could not observe any significant differences between the relative velocities in the presence of wildtype FUS-GFP and FUS-P525L-GFP versus our controls (buffer only and BSA, Figure 3D). In line with this, no significant protein aggregation over time was evident for both FUS variants (Appendix A). Hence, we attribute the slight velocity decline over time to the aging of the assay rather than to inhibitory effects by (aggregated) FUS.

### 2.3. Cell Lysates from Neurons Expressing FUS-P525L-GFP Do Not Inhibit Kinesin-1 Motility

While we found that recombinant wildtype FUS-GFP or FUS-P525L-GFP alone did not interfere directly with kinesin-1-dependent microtubule gliding, we asked whether other cytosolic factors (e.g., the presence of additional motor and microtubule effector proteins [48,49] or the trapping of MAPs in aggregates [36]) may have an impact together with FUS. To answer this question, we tested whole-cell lysates from CRISPR/Cas9-engineered isogenic iPSCs-derived spinal motor neurons expressing GFP-labeled versions of wildtype FUS and FUS-P525L for effects on kinesin-1 motility (Figure 4A). The generation of the lines and their thorough phenotyping was described previously [27,50]. Previous studies showed that increasing concentrations of cell lysates can decrease kinesin‑1‑dependent microtubule gliding [51]. However, when added to the assay at final total protein concentrations of 50, 80, 110, and 140 ng/µL, the microtubule gliding velocities were—irrespective of the expressed FUS variant—not significantly affected in comparison to control experiments with BSA (Figure 4B).

Next we performed a western blot on cell lysates expressing either wildtype FUS‑GFP or FUS‑P525L‑GFP (Appendix A). Endogenous levels of either FUS variant within cell lysates were in the range comparable to 2.5 and 5nm recombinant FUS‑GFP variants. The latter have been added to our modified kinesin‑1‑dependent microtubule gliding assay including much higher concentrations (5–5000 nM, Figure 3), which clearly suggests that FUS has no direct effect on kinesin-1-dependend microtubule gliding in “physiological” concentrations within neurons. In addition, to exclude a decreased assay sensitivity in the presence of neuronal whole‑cell lysates, we re-evaluated the assay performance by titrating low amounts (i.e., 1–100 nM final concentration) of recombinant human 2N4R tau-GFP into the lysates (at a total protein concentration of 80 ng/µL), asking whether we still see any inhibitory effects on microtubule gliding velocities below 50 nM tau. In contrast to our experiments with recombinant 2N4R tau-GFP alone, the relative gliding microtubule velocity started to decrease already in the presence of 5 nM 2N4R tau-GFP, but also completely ceased at 50 nM 2N4R tau-GFP, irrespective of the expressed FUS variant in the cell lysates (Figure 4C). We hence conclude that, even in the presence of whole‑cell lysates, our assay is still sufficiently sensitive to inhibitors of kinesin-1 motility and that neither lysates from healthy motor neurons nor those from ALS-patient-derived motor neurons negatively affect kinesin-1 motility.

## 3. Discussion

Axonal transport defects are characteristic to virtually every neurodegenerative disease. Hence, a robust assay to study and manipulate the axonal transport machinery is of general value. Here, we report adjustments to a classic kinesin-1-dependent microtubule gliding motility assay, improving its robustness, reproducibility and sensitivity so that it can detect low nanomolar amounts of factors that interfere with kinesin motility even in high protein abundance conditions (i.e., whole‑cell lysates). Towards this end, we avoided phosphatase inhibitor cocktails that interfere with microtubule gliding (Appendix A), presumably by affecting the ATPase activity of motor proteins [52,53,54]. Instead, we used β‑glycerophosphate serving as an alternative phosphatase substrate to preserve the phosphorylation state of proteins in whole‑cell lysates. Unfortunately, however, high amounts of the negatively charged β-glycerophosphate interfere with motor-microtubule binding (Appendix A), most likely by interfering with the electrostatic interactions between the lattice-exposed acidic C-termini of alpha/beta tubulin (E-hook) and the positively charged K-loop of kinesin-1 [55]. In our optimized assay conditions we limited the resulting increase in microtubule detachment from the surface-bound kinesin motors by the addition of methylcellulose. Methylcellulose increases the viscosity of the assay buffer, which prevents detached microtubules from escaping into solution and facilitates their efficient recapture by surface bound motors [56] (Appendix A). Moreover, we took into account that the motility of molecular motors is highly sensitive to temperature fluctuations [57]. To improve the reproducibility of our motility measurements, we hence minimized temperature fluctuations using an objective heater and an additional Peltier element firmly attached to the flow channels. This setup allowed us to control the flow channel temperature with an accuracy of 0.1 degree Celsius. Our optimized kinesin-1-dependent microtubule gliding motility assay produces highly reproducible results at a high sensitivity: consistent with previous studies, we found that kinesin-1-dependent microtubule gliding is inhibited at a concentration of full-length 2N4R tau-GFP of about 50 nM and higher (Figure 2B) [58,59]. The same sensitivity was determined in the presence of whole‑cell lysates (Figure 4C), underlining the robustness of our assay.

Axonal transport has been shown to be dramatically impaired in FUS-ALS patient-derived motor neurons [27,29,50]; however, the underlying mechanism is still puzzling. We used our optimized kinesin-1-dependent microtubule gliding motility assay to test whether ALS-associated FUS variants directly affect kinesin-1-driven transport e.g., by acting as a road block like tau. We show that neither recombinant wildtype FUS-GFP nor FUS-P525L-GFP nor cell lysates from motor neurons expressing those FUS variants reduce the microtubule gliding velocity over a wide range of concentrations. Furthermore and in contrast to tau, wildtype and mutant FUS did not directly bind to microtubules (Figure 3B). The fact that the velocities of microtubule gliding slightly decreased over the course of three hours irrespective of the presence/absence of additional proteins is likely due to an overall decrease in assay performance (“aging”) of the transport machinery (i.e., caused by changes in pH, depletion of ATP or oxidative damage due to depletion of oxygen scavenging components). Hence we conclude that mislocalized cytoplasmic FUS (i.e., FUS-P525L) does neither directly nor indirectly interfere with kinesin-1-dependent (anterograde) transport.

Regarding the study of possible indirect effects of FUS on kinesin-1-dependent transport, the results of our kinesin-1-dependent microtubule gliding motility assay in the presence of motor neuron lysates have some limitations which might arose due to the loss of compartmentalization during cell lysis. For example, current models suggest that pathogenic FUS variants may indirectly impair axonal transport of mitochondria and endoplasmic reticulum (ER) vesicles [16,27,29], e.g., by promoting mitochondrial damage [60], pathological protein aggregation [61,62], or by impairing DNA repair mechanisms resulting in altered protein expression [27,62]. Damaged mitochondria cease to synthesize ATP, causing a nearby drop in axonal ATP concentration, which locally impairs ATP-dependent transport. These local effects, however, are lost in whole cell lysates and therefore cannot be reproduced in our protein-enriched kinesin-1-dependent microtubule gliding motility assay. Additionally, the ATP added to our assay mix will mask potential ATP depletion effects in ALS-associated motor neurons or lysates thereof. Complementary experimental approaches, for example the tracking of individual, cargo-binding-deficient, exogenous kinesin-1 motors in living neurons, will be needed to answer those questions.

Moreover, NLS-mutations that prevent the nuclear import of the transcription factor FUS cause altered expression levels of its target genes [63]. The expression levels of the major α-tubulin deacetylase HDAC6, for example, are elevated in patient-derived motor neurons expressing disease-associated NLS variants of FUS, resulting in aberrant acetylation and hence detrimental stabilization of microtubules, ultimately causing axonal transport defects [29,64]. The NLS-mutant FUS variants (P525L, R521C, R495X) also form inclusions that sequester, amongst others, kinesin-1 mRNA and protein, which targets the tubulin carboxypeptidase enzyme onto specific microtubules. Its depletion hence leads to inhibition of microtubule detyrosination [36], suggesting that there are different subsets of pathologically altered microtubules in FUS-ALS. However, the microtubules used in our assay were assembled from purified porcine (neuronal) tubulin of unknown posttranslational modification status. Hence, our assay cannot be used to detect changes in endogenous microtubule modifications potentially caused by mislocalized FUS in vivo. Such modifications can be investigated by gently isolating the endogenous cytoskeleton of neurons, a process which preserves at least the acetylation and tyrosination state of microtubules [65], and the kinesin-1 motility studied by subsequent single-molecule imaging on these microtubules in the presence of FUS variants. Nevertheless, FUS stress granule appearance is a more downstream mechanisms appearing either late in iPSC-derived neuronal cell cultures or upon induction of cell stress [27]. We used cell lysates after three weeks of maturation, knowing that at this time point axonal transport defects are present while cytoplasmic accumulation is not in the absence of additional stress induction, in order to address whether the observed axonal transport deficiency is kinesin-1 dependent. Thus we do not believe that sequestered kinesin-1 is the cause for an early impairment of axonal transport as observed in cell culture assays [27]. Finally, pathogenic FUS variants are known to form insoluble aggregates [22,66], which may sterically hinder or block cargo from being transported through the axon [67]. Even in the unlikely case that endogenous FUS-P525L aggregates were initially present in motor neuron lysates at the time point of analysis [27], we likely missed them out in our motility assay, as they were presumably lost at the cell debris clearance step during lysate preparation. Of note, at the given conditions of our assay, however, FUS aggregation did also not occur in the experiments using recombinant FUS over the course of three hours even at high concentrations of recombinant wildtype FUS-GFP and FUS-P525L-GFP (Appendix A).

Interestingly, a recent paper suggests an oligogenic model for ALS after variants of tank-binding kinase 1 (TBK1) were found in patients also expressing ALS-associated FUS variants [68] and both proteins individually have been linked to ALS as well as FTD [69]. At least in such cases of oligonenic architecture, defects in axonal transport might not be caused by FUS, but by other genetically altered variants. This is however unlikely in our setup having used isogenic cell lines in which only the FUS mutation differs between the cells.

## 4. Materials and Methods

### 4.1. NPC Line Generation, Cell Culture and Motor Neuron Differentiation

Neural precursor cell (NPC) lines were derived from isogenic iPSC lines carrying either wildtype FUS-GFP or FUS-P525L-GFP, which was previously inserted by CRISPR/Cas9-mediated genome editing via homology directed repair [27,70]. All procedures were in accordance with the Helsinki convention and approved by the Ethical Committee of the University of Dresden (EK45022009; EK393122012). NPC lines were generated using established protocols [71]. NPCs were maintained in N2B27 medium (48.75% DMEM/F12, 48.75% Neurobasal^®^ Medium, 1% penicillin/streptomycin, 1% B27 supplement without vitamin A, and 0.5 % N_2_ supplement, all purchased from Thermo Fisher Scientific, Waltham, MA, USA and Life Technologies Corporation, Carlsbad, CA, USA) on Matrigel^TM^ (BD Bioscience, Erembodegem, Belgium) coated plates at standard cell culture conditions (37 °C, 5% CO_2_). To induce motor neuron differentiation, NPCs were cultured (patterned) in N2B27 medium supplemented with 10 ng/mL recombinant human brain-derived neurotrophic factor (rhBDNF, Promega GmbH, Mannheim, Germany), 0.2 mM ascorbic acid (AA, Sigma, St. Louis, MO, USA), 1 µM retinoic acid (RA, Sigma, St. Louis, MO, USA), 0.5 µM Smoothened Agonist (SAG, Cayman, Ann Arbor, MI, USA), and 1 ng/mL recombinant human glial cell line-derived neurotrophic factor (rhGDNF, Sgma, St. Louis, MO, USA). After a final split of these patterned neurons and subsequent seeding onto plates coated with 15% Poly-L-Ornithine (PLO, Sigma, St. Louis, MO, USA) and 0.005 mg/mL Laminin (Roche, Basel, Switzerland), cells were kept in culture for 21 days in N2B27 medium containing 0.1 mM dibutyryl-cyclic adenosine monophosphate (dbcAMP, Sigma, St. Louis, MO, USA), 2 ng/mL rhBDNF, 0.2 mM AA, 1 ng/mL TGFβ-3, 2 ng/mL rhGDNF, and 5 ng/mL Activin A (Biomol GmbH, Hamburg, Germany) on the first day of culture.

### 4.2. Preparation of Cell Lysates and Protein Concentration Determination

On day 21 of maturation, cells were gently washed with lysis buffer consisting of PBS (Life Technologies Corporation, Carlsbad, CA, USA ) supplemented with 1x protease inhibitor (Halt™, Thermo Scientific™, Waltham, MA, USA), and 10 mM β-glycerophosphate (Sigma, St. Louis, MO, USA). Cells were then scraped off the flask surface, resuspended in 2 mL lysis buffer and pelleted at 14,000× *g* and 4 °C for 6 min. The cell pellet was resuspended in 1 mL lysis buffer and pulled at least 10 times through a needle (0.4 × 19 mm; 27G ¾). The homogenized cell suspension was cleared at 14,000× *g* and 4 °C for 6 min. The final lysates were supplemented with 10% glycerol (MP biomedicals, Santa Ana, CA, USA), aliquoted, snap frozen in liquid nitrogen, and stored at −80 °C. Protein concentration has been determined using the Pierce™ BCA Protein Assay Kit (Thermo Scientific™, Waltham, MA, USA) according to the manufacturer’s instructions.

### 4.3. Expression and Purification of Tubulin and Kinesin-1

Tubulin was purified from porcine brain (obtained from Vorwerk Podemus, Dresden, Germany) using previously established protocols [72] and fluorescently labeled using 5-(and 6-) carboxytetramethylrhodamine, succinimidyl ester (5(6)-TAMRA,SE) mixed isomers (C1171, Invitrogen, Carlsbad, CA, USA). Full length *Drosophila melanogaster* kinesin-1 was expressed in insect cells and purified as previously described [73].

### 4.4. Protein-Enriched Microtubule Gliding Motility Assay

Kinesin-microtubule gliding motility assays were performed as described previously [37] with slight modifications. In brief, microtubules were polymerized from 40 μM rhodamine-labeled tubulin in BRB80 (80 mM PIPES, 1 mM EGTA, 1 mM MgCl_2_), 4 mM MgCl_2_, 1 mM Mg-GTP, and 5% DMSO at 37 °C for 15 min. Unless otherwise stated, all chemical were purchased from all Sigma, St. Louis, MO, USA. Polymerized microtubules were then diluted 100-fold in BRB80 containing 10 μM taxol and sheared twice using a Kel-F Hub needle (90530, Hamilton, Bonaduz, Switzerland) and 1 mL Luer-Lok^TM^ syringe (BD Bioscience, Erembodegem, Belgium).

Flow channels assembled from two glass cover slips (volume of about 18 mm × 2 mm × 0.1 mm, Thermo Scientific™, Waltham, MA, USA) were first perfused with casein-containing solution (0.5 mg/mL) in PBS and left to adsorb for 5 min in order to reduce non-specific protein adsorption to the glass surfaces. Next, 20 μL of a PBS solution containing *drosophila* full length kinesin-1 (4 μg/mL), casein (0.2 mg/mL), 1 mM ATP, and 10 mM DTT was flushed through the channels in order for kinesin‑1 to attach to the glass surfaces. After 5 min, flow channels were loaded with rhodamine-labeled taxol-stabilized microtubules, diluted 10-fold in standard motility buffer (PBS supplied with 10% glycerol, 10 mM β-glycerophosphate, 0.3% methylcellulose, 1 mM ATP, 20 mM D-glucose, 10 μg/mL glucose oxidase, 10 μg/mL catalase, 10 mM DTT, and 10 μM taxol). Flow channels were fixed on a custom-build Peltier element for temperature control during imaging.

Imaging was performed using a Nikon Eclipse Ti fluorescence microscope equipped with a Perfect Focus System, a 1.49 PlanApo 100× oil immersion objective heated to 30 °C, and an iXon Ultra back-illuminated EMCCD camera in conjunction with Nikon (Tokyo, Japan) NIS-Elements imaging software. Rhodamine-labeled microtubules were observed by widefield epi-fluorescence, excited with a 550 nm LED lamp. Time-lapse movies were recorded for 10 frames (at an acquisition rate of one frame per second) with exposure times of 100 ms at ten different positions in the flow channel (standard condition). Thereafter, the flow channels were perfused with recombinant tau-GFP (kindly provided by Markus Braun and Zdeněk Lánský, Institute of Biotechnology of the Czech Academy of Sciences, Prague, Czech Republic), recombinant wildtype FUS-GFP or FUS-P525L-GFP (kindly provided by J. Wang, Max Planck Institute of Molecular Cell Biology and Genetics, Dresden, Germany), BSA (Sigma, St. Louis, MO, USA), or the respective cell lysate, each supplemented with 0.3% methylcellulose, 20 mM D-glucose, 20 μg/mL glucose oxidase, 10 μg/mL catalase, 10 mM DTT, and 1 mM ATP. The same ten positions were imaged 5 min after protein addition as described before. To image tau-GFP and FUS-GFP, single frames of all ten positions were captured with a 470 nm LED lamp after protein addition, immediately before acquiring time-lapse movies of the rhodamine-labeled microtubules.

### 4.5. Microtubule Pull-Down Assay

Microtubules were polymerized from 40 μM unlabeled tubulin in BRB80 (80 mM PIPES, 1 mM EGTA, 1 mM MgCl_2_), 4 mM MgCl_2_, 1 mM Mg-GTP, and 5% DMSO at 37 °C for 20 min. Polymerized microtubules were cleared by centrifugation at 17,000× *g* at 21 °C for 30 min. and then diluted in 114 µL PBST (PBS containing 10 μM taxol). Polymerized microtubules were then incubated with 500 nM tau, wildtype FUS or FUS-P525L-GFP, respectively, in PBST for 20 min at room temperature. Samples were then centrifuged at 45,000× *g* and 21 °C for 20 min, after which the supernatant was separated from the microtubule-containing pellet. SDS loading buffer (Thermo Scientific™, Waltham, MA, USA), supplemented with 0.1 M DTT, was mixed 1:4 with each sample and boiled at 95 °C for 5 min. Samples were loaded onto a 4–12% Bis-Tris gel (Thermo Scientific™, Waltham, MA, USA),. The gel was run in 1× MOPS buffer (Thermo Scientific™, Waltham, MA, USA) at 180 V, 85 W, and 60 mA until all protein had passed to the end of the gel (at least one hour). Gels were then stained with Coomassie SimplyBlue SafeStain (Thermo Scientific™, Waltham, MA, USA) for one hour and destained in ddH2O until bands were clearly visible (at least one hour). Gels were then scanned with an Azure c600 Gel Imaging System (Azure Biosystems Inc., Dublin, CA, USA).

### 4.6. Western Blot

Samples for western blot analysis were mixed with NuPage loading buffer (Thermo Scientific™, Waltham, MA, USA) in a 3:4 ratio, heated to 95 °C for 5 min and loaded onto a 4–12% Bis-Tris SDS-PAGE gel (Thermo Scientific™, Waltham, MA, USA). The gel was run in 1 × MOPS buffer (Thermo Scientific™, Waltham, MA, USA) for at least one hour at 140 V, 85 W, and 60 mA. Following electrophoresis, gels were kept in transfer buffer supplemented with 0.1% SDS for 8 min to increase protein release from the gel during blotting. In contrast, a PVDF membrane was soaked in transfer buffer containing 6.7% methanol prior to blotting in order to activate the membrane and increase protein retention. iBlot transfer stacks (Thermo Scientific™, Waltham, MA, USA) were assembled according to the manufacturer’s instructions in the iBlot transfer system and blotted for 7 min with 19 V. After blotting, membranes were transferred into 50 mL falcons and blocked for one hour at room temperature in PBS supplemented with 0.05% Tween20 and 5% milk powder. Membranes were then incubated with primary antibodies over night at 4 °C and the next day with horseradish peroxidase-coupled secondary antibodies for one hour at RT. Chemiluminescent detection was achieved by applying Amersham ECL Prime (GE Healthcare Life Sciences, Marlborough, MA, USA) substrate according to the manufacturer’s instructions. Imaging was immediately performed after substrate application on an LAS3000 imager (Fujifilm, Tokyo, Japan). To determine the concentration of FUS variants in whole cell lysates, recombinant wildtype FUS-GFP of known concentrations was loaded on SDS PAGE gels in combination with whole cell lysates. After detection, membranes were stripped by incubation in stripping buffer (3 g glycine, 0.1% SDS, 1% Tween20, pH 2.2, all Sigma, St. Louis, MO, USA) twice at room temperature for 10 min each, two subsequent washing steps with PBS for 10 min each, and two final washing steps with PBS supplemented with 0.05% Tween20 for 5 min each. After this procedure, membranes were again blocked and incubated with respective primary and secondary antibodies as described above in order to normalize endogenous FUS levels to the housekeeping gene β-actin.

### 4.7. Data Analysis and Statistics

Microtubule gliding velocities were derived from the time-lapse movies using an automated MATLAB (The MathWorks, Inc., Natick, MA, USA) script based on a microtubule tip-tracking algorithm developed in-house [45,46]. In brief, for each frame, the position of a microtubule tip was localized with sub-pixel resolution and compared to its position in the subsequent frame, thereby determining the frame-to-frame velocity. The frame-to-frame velocities of all microtubules imaged under one condition (i.e., in presence of tau-GFP, FUS-GFP, BSA, or cell lysate) in one flow channel (i.e., data from ten frames at ten different positions) were pooled and the median velocity was determined. For better comparison, relative velocities were derived by dividing these median velocities by the median velocities under standard condition in the same channel (i.e., derived in the same manner in presence of standard motility buffer before application of the protein mixes). Averages of at least three independent experiments ± standard deviation (STD) are shown in Figure 2C, Figure 3C,D, Figure 4B,C and Appendix A.

Significance (see Appendix A for a statistical analysis of all acquired data) was determined using two-way ANOVA and Tukey’s multiple comparison post-hoc test in GraphPad Prism (Version 7, GraphPad Software Inc., La Jolla, CA, USA). Kymographs were created using the Kymograph plugin for ImageJ created by J. Rietdorf (FMI, Basel, Switzerland) and A. Seitz (EMBL, Heidelberg, Germany), while overlay images were generated using MetaMorph (Molecular Devices, Sunnyvale, CA, USA, Version 7.8.4.0).

## 5. Conclusions

Taken together, we established an optimized kinesin-1-dependent microtubule gliding motility assay that tolerates the presence of whole‑cell lysates and provides sensitive, highly reproducible and robust readouts of kinesin-1 motility. It therefore constitutes a useful tool to study the underlying mechanism of axonal transport defects in neurodegenerative diseases in vitro. Using this assay, we showed that neither wildtype FUS nor the FUS-P525L variants directly bind to Tau or directly interfere with kinesin-1 motility. Within the limits of the assay described above, we could also show that neither FUS variant affects kinesin-1 motility indirectly via other factors present in motor neuron lysates. We conclude that the axonal transport defects observed in various FUS‑ALS models are not caused by direct interaction with microtubule-dependent kinesin-1 motility. The precise mechanisms by which FUS may indirectly affect axonal transport (e.g., by aggregating or altering microtubule posttranslational modification) remain to be elucidated in future studies.

## Figures and Tables

**Figure 1 ijms-22-02422-f001:**
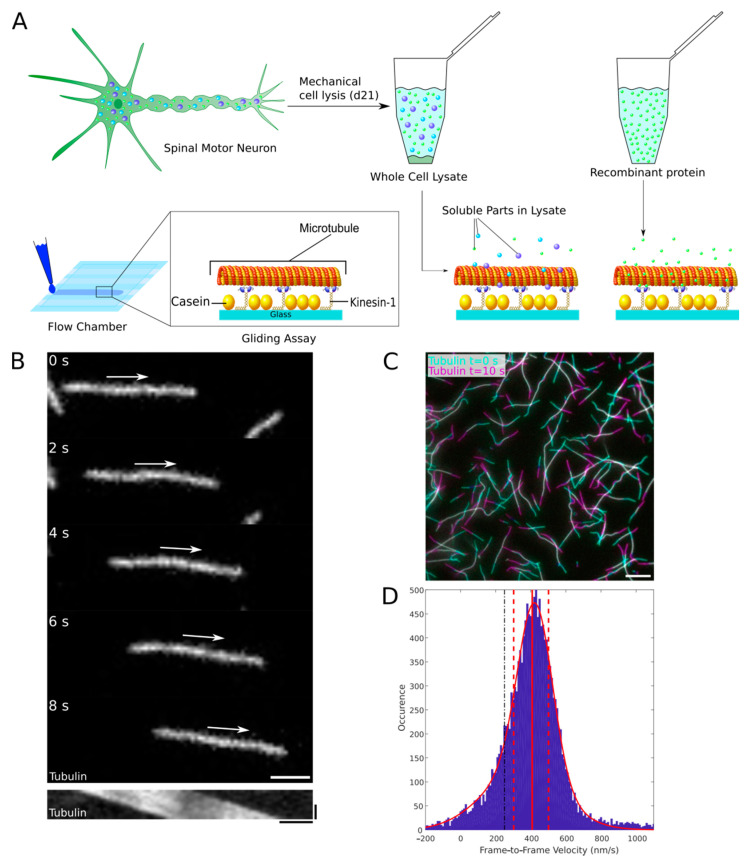
Reconstituting axonal transport in vitro. (**A**) ALS-Patient-specific and isogenic control iPSCs were differentiated to form spinal motor neurons. After 21 days of maturation, cells were mechanically lysed and the soluble fractions of the cell lysates were collected by centrifugation, further referred to as whole-cell lysates. Either lysates or recombinant protein was then applied to a kinesin-1-dependent microtubule gliding motility assay in self-assembled glass flow chambers. (**B**) Top panels: Microtubules were imaged using epifluorescence microscopy for 10 s (acquisition rate of one frame per second, exposure time of 100 ms) at ten different fields of view (Micrographs only show a small part of the field of view, Scale bar = 5 µm, white arrow indicates direction of movement). Bottom panel: Representative kymograph showing the kinesin-1-dependent movement of the above microtubule. Horizontal scale bar = 2 µm, vertical = 5 s. (**C**) Overlay image illustrating the kinesin-1-dependent gliding of rhodamine-labeled microtubules in the presence of wildtype FUS-GFP cell lysate. Microtubule positions at time points t = 0 s (cyan) and t = 10 s (magenta) are overlaid. White color in the overlay indicates positional overlap at both time points whereas cyan and magenta colors indicate the moved distances. Scale bar = 10 µm. (**D**) Frame-to-frame velocities were derived using an automated MATLAB script. Data plotted in a histogram with Gaussian fit for inspection (red curve, red solid and dashed lines mark the median as well as the 25th and 75th percentiles, respectively).

**Figure 2 ijms-22-02422-f002:**
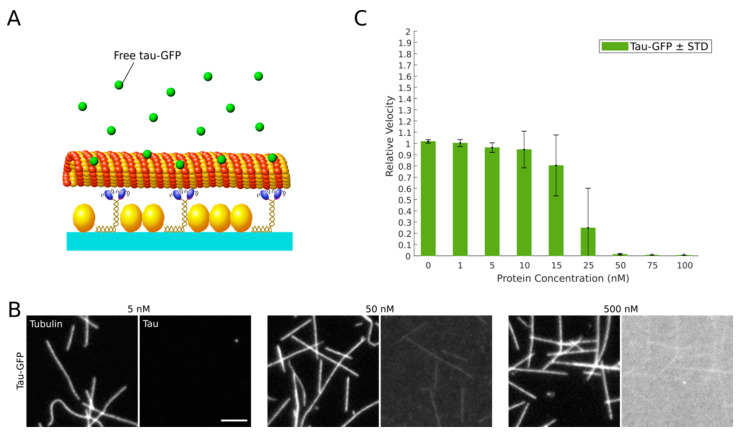
Tau-GFP inhibits microtubule gliding on kinesin-1 motors in a concentration-dependent manner. (**A**) Recombinant human 2N4R tau-GFP was applied to our optimized microtubule gliding motility assay. (**B**) Binding of tau-GFP (right panels) to taxol-stabilized rhodamine-labeled microtubules (left panels) at the indicated concentrations. Scale bar = 5 µm. (**C**) Relative kinesin-1-dependent microtubule gliding velocities (see Materials and Methods) at the indicated tau-GFP concentrations. Significance levels are indicated in Appendix A.

**Figure 3 ijms-22-02422-f003:**
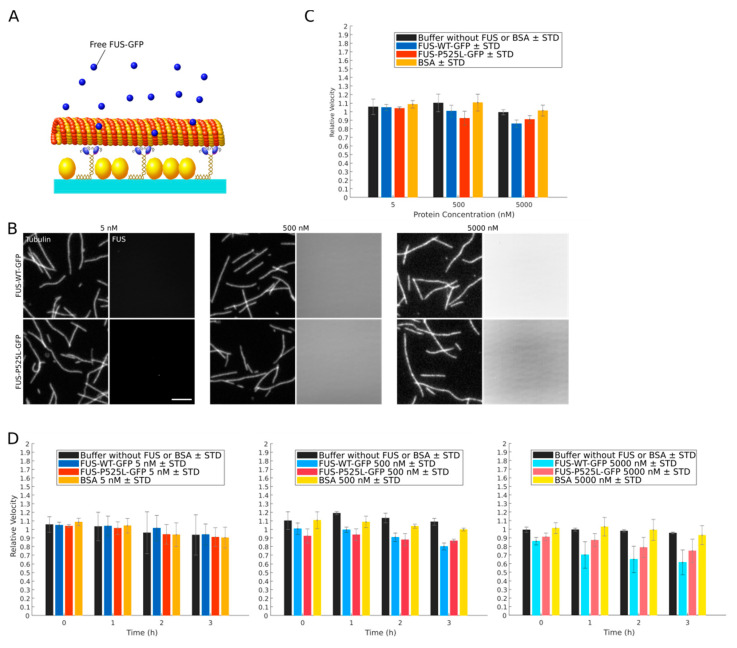
Recombinant wildtype FUS-GFP and FUS-P525L-GFP do not inhibit microtubule gliding on kinesin-1 motors. (**A**) Recombinant human wildtype FUS-GFP or FUS-P525L-GFP was applied to a kinesin-1-dependent microtubule gliding assay. (**B**) Binding of FUS-GFP (right panels) to taxol-stabilized rhodamine-labeled microtubules (left panels) at the indicated concentrations. Scale bar = 5 µm. (**C**) Relative kinesin-1-dependent microtubule gliding velocities (see Materials and Methods) at the indicated FUS or control protein (BSA) concentrations. Significance levels are indicated in Appendix A. (**D**) Relative kinesin-1-dependent microtubule gliding velocities at three concentrations of wildtype FUS-GFP, FUS-P525L-GFP, or BSA over time. Significance levels are indicated in Appendix A.

**Figure 4 ijms-22-02422-f004:**
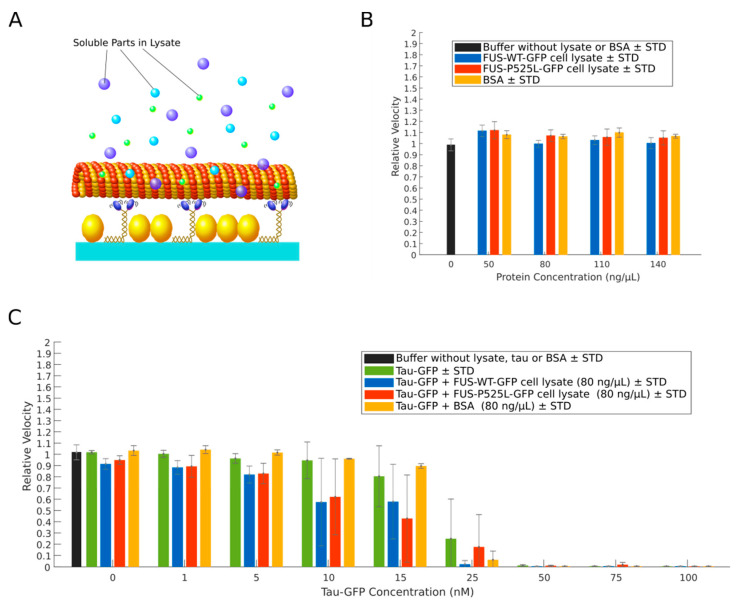
Cell lysates of motor neurons do not inhibit microtubule gliding on kinesin-1 motors. (**A**) Wildtype FUS‑GFP or FUS‑P525L‑GFP cell lysates were applied to a kinesin-1-dependent microtubule gliding motility assay.(**B**) Relative kinesin‑1‑dependent microtubule gliding velocities (see Materials and Methods) at indicated concentrations of total protein in wildtype FUS-GFP and FUS-P525L-GFP cell lysates or protein control (BSA). Significance levels are indicated in Appendix A. (**C**) Relative kinesin-1-dependent microtubule gliding velocities in wildtype FUS-GFP, FUS-P525L-GFP cell lysates or protein control (BSA) at a concentration of 80 ng/µL supplemented with human 2N4R tau-GFP at the indicated concentrations. Data with 2N4R tau-GFP only replotted from Figure 2C for comparison. Significance levels are indicated in Appendix A.

## Data Availability

All data are made available within this manuscript.

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
