# Peer review of "The ALS-Associated FUS (P525L) Variant Does Not Directly Interfere with Microtubule-Dependent Kinesin-1 Motility"

_ijms, 2021, doi:10.3390/ijms22052422_

Round 1

Reviewer 1 Report

In this manuscript, Seifert et al. have firstly optimised/adapted a previously described kinesin-1-dependent in vitro microtubule gliding assay. They then used the adapted assay to determine if a human mutant form of the fused-in-sarcoma (FUS) gene (P525L) (recombinant and expressed in iPSC-derived motor neurons) impacted kinesin-dependent motility of microtubules. The authors observed that the mutant FUS protein did not impact microtubule motility compared to a recombinant tau construct used as a control. The authors therefore conclude that while the human FUS P525L mutation leads to axonal transport defects, as reported by several other research groups, that these are not directly due to the mutant protein itself and perhaps is indirectly caused by non-FUS molecular effectors. While the results as they are presented are interesting, they are drawn from experiment paradigms that require additional controls and/or methodologies to support the experimental approaches used and subsequent conclusions.

Comments/suggestions (minor and major):

  1. In some instances, there are no spaces between sentences.
  2. While the authors have used recombinant full-length human tau as a control for antagonising activity, have the authors used validated drugs (e.g. eribulin, vincristine, paclitaxel and ixabepilone https://www.ncbi.nlm.nih.gov/pmc/articles/PMC4169189/) to validate their adapted assay further?
  3. While the authors mention that the CRISPR/CAS9 iPSC-derived motor neurons have previously published and described, can the authors provide evidence and validate that in their hands, the iPSC-derived motor neurons had the expected phenotype?
  4. Furthermore, it seems a bot illogical to introduce the iPSCs in the first section of the Results when as an experimental model, they do not appear until Figure 4 (or maybe they do, but it is not apparent).
  5. Figure 2B: The microscopy images of tau-GFP appear to be at different exposure settings and should therefore be normalised. An anti-GFP antibody can be used to amplify the signal. Furthermore, alternative approaches such as immunoprecipitation and proximity-ligation assays could have been used to confirm/validate binding of recombinant protein to microtubules. This applies to all subsequent figures with similar experimental approaches (3B, Supplementary Figure 2) .
  6. Figure 4: Total protein was quantified but was total FUS (WT or mutant) quantified and how did it compare to the amount of recombinant 2N4R tau-GFP added to the assays?
  7. Materials & Methods: First paragraph needs to be deleted as it part of the "author guidelines".

Author Response

Reviewer 1:

In this manuscript, Seifert et al. have firstly optimised/adapted a previously described kinesin-1-dependent in vitro microtubule gliding assay. They then used the adapted assay to determine if a human mutant form of the fused-in-sarcoma (FUS) gene (P525L) (recombinant and expressed in iPSC-derived motor neurons) impacted kinesin-dependent motility of microtubules. The authors observed that the mutant FUS protein did not impact microtubule motility compared to a recombinant tau construct used as a control. The authors therefore conclude that while the human FUS P525L mutation leads to axonal transport defects, as reported by several other research groups, that these are not directly due to the mutant protein itself and perhaps is indirectly caused by non-FUS molecular effectors. While the results as they are presented are interesting, they are drawn from experiment paradigms that require additional controls and/or methodologies to support the experimental approaches used and subsequent conclusions.

Response: We appreciate this overall positive review. We got the impression that our message was maybe a bit mistaken. We only wanted to conclude that while the human FUS P525L mutation leads to axonal transport defects, as reported by several other research groups and ourselves (DOI: 10.1038/s41467-017-02299-1), they are not due to a direct interaction between the FUS protein and tubulin/Kinesin-1 themselves, but perhaps is indirectly caused by either indirect effects of FUS itself (e.g. transcriptional regulation (DOI: 10.1074/jbc.273.43.27761), splicing regulation (DOI: 10.3389/fmolb.2018.00044), binding/sequestration of tubulin/kinesin-1 interaction partners (DOI: 10.3389/fncel.2015.00423 and 10.1016/j.neuron.2004.07.022) or non-FUS molecular effectors. We aimed to clarify this is the revised version of the manuscript.

Comments/suggestions (minor and major):

1. In some instances, there are no spaces between sentences.

Response: We corrected this.

2. While the authors have used recombinant full-length human tau as a control for antagonising activity, have the authors used validated drugs (e.g. eribulin, vincristine, paclitaxel and ixabepilone https://www.ncbi.nlm.nih.gov/pmc/articles/PMC4169189/) to validate their adapted assay further?

Response: We appreciate this comment. The mentioned drugs are all well described drugs to interfere with microtubules, e.g. vincristine und ixabepilone slow down microtubule gliding in gliding assays (DOI: 10.1016/j.neuro.2013.05.008). However, we only wanted to proof whether the gliding assay does what it should or react as expected and thus do believe that the results shown with tau protein are sufficiently convincing to proof that tau slows down microtubule gliding assay and decorates microtubules already at nanomolar concentrations. In addition, tau is a more physiological influencer of microtubule gliding motility and we expect microtubule-binding proteins to behave similar to tau rather than to artificial drugs.

3. While the authors mention that the CRISPR/CAS9 iPSC-derived motor neurons have previously published and described, can the authors provide evidence and validate that in their hands, the iPSC-derived motor neurons had the expected phenotype?

Response: As mentioned in the manuscript, we generated and published these lines ourselves and thus can say that – as shown in the published papers (DOI: 10.1038/s41467-017-02299-1) – they do show the expected phenotypes in our hands.

4. Furthermore, it seems a bit illogical to introduce the iPSCs in the first section of the Results when as an experimental model, they do not appear until Figure 4 (or maybe they do, but it is not apparent).

Response: On a first view we do understand this point, however since all analysis depend on this assay and since the assay conditions need to be the same/similar between recombinant proteins and cell lysates, we do believe it still makes sense to have this part here while in parallel improved it.

5. Figure 2B: The microscopy images of tau-GFP appear to be at different exposure settings and should therefore be normalised. An anti-GFP antibody can be used to amplify the signal. Furthermore, alternative approaches such as immunoprecipitation and proximity-ligation assays could have been used to confirm/validate binding of recombinant protein to microtubules. This applies to all subsequent figures with similar experimental approaches (3B, Supplementary Figure 2).

Response: The microscopy images are taken with the same exposure settings and are scaled equally, which makes the background appear much brighter in the presence of higher amounts of tau-GFP. A microtubule pull-down assay has been performed and included in the revised version of the manuscript to confirm that FUS variants are not binding to microtubules (Figure S2).

6. Figure 4: Total protein was quantified but was total FUS (WT or mutant) quantified and how did it compare to the amount of recombinant 2N4R tau-GFP added to the assays?

Response: We appreciate the suggestion. We included a supplementary figure in the manuscript showing the endogenous protein levels of FUS‑GFP variants expressed in whole cell lysates,determined by western blot in comparison to the well defined concentrations of exogenously added recombinant FUS (Figure S3). Externally added tau-GFP was in the same range but titrated to also higher concentrations, as it was done in parallel with recombinant FUS protein.

7. Materials & Methods: First paragraph needs to be deleted as it part of the "author guidelines".

 Response: We apologize and of course did delete the paragraph.

Reviewer 2 Report

In their paper Seifert et al. examine the possible interfering effect of FUS and FUS variant (P525L) on kinesin-1 motility, which may explain the impaired axonal transport in ALS patients. Using an improved gliding assay the authors test whether FUS protein affects kinesin-1 directly, by inhibiting kinesin-1 motility, or indirectly, by roadblock mechanism. They show that both WT and mutant have no effect, direct or indirect, on gliding velocity of the motor protein.

The paper is well written and experiments are well performed. Yet, I suggest the authors to add a pelleting assay that will support their claim that FUS doesn’t bind to MTs. While the significance of the study is a little bit limited, I strongly support the publication of “negative” results as I think it will be beneficial for the scientific community.

Minor comments:

P8- typo mistake - "optimizedkinesin-1"

P9 – 1st paragraph of the Material and Methods should be removed

The text in the chart legends in figures 2-4 should be larger

Author Response

Reviewer 2:

In their paper Seifert et al. examine the possible interfering effect of FUS and FUS variant (P525L) on kinesin-1 motility, which may explain the impaired axonal transport in ALS patients. Using an improved gliding assay the authors test whether FUS protein affects kinesin-1 directly, by inhibiting kinesin-1 motility, or indirectly, by roadblock mechanism. They show that both WT and mutant have no effect, direct or indirect, on gliding velocity of the motor protein.

The paper is well written and experiments are well performed. Yet, I suggest the authors to add a pelleting assay that will support their claim that FUS doesn’t bind to MTs. While the significance of the study is a little bit limited, I strongly support the publication of “negative” results as I think it will be beneficial for the scientific community.

Response: We deeply appreciate this review! A microtubule pull-down assay has been performed and included in the manuscript to confirm that FUS variants are not binding to microtubules (Figure S2).

Minor comments:

P8- typo mistake - "optimizedkinesin-1"

P9 – 1st paragraph of the Material and Methods should be removed

The text in the chart legends in figures 2-4 should be larger

Response: We corrected these.

Reviewer 3 Report

Seifert et al. aimed at investigating whether there is any impact of wildtype FUS and FUS-P525L on anterograde axonal transport, exemplified by in vitro reconstituted microtubule gliding motility assays on surfaces coated with kinesin-1 motors. The Authors used whole-cell lysates obtained from Amyotrophic Lateral Sclerosis (ALS)-patient-specific iPSC-derived spinal motor neurons which endogenously express these FUS variants. They hypothesized that mislocalized pathogenic FUS variants might causally contribute to the observed axonal transport phenotypes of ALS by direct interaction of free FUS with the transport machinery (e.g., directly inhibiting motor proteins or acting as road blocks on microtubules). The results showed that neither recombinant wildtype FUS-GFP nor FUS-P525L-GFP nor cell lysates from motor neurons expressing those FUS variants reduce the microtubule gliding velocity over a wide range of concentrations. Furthermore and in contrast to Tau, wildtype and mutant FUS did not directly bind to microtubules. Hence the Authors concluded that mislocalized cytoplasmic FUS (i.e. FUSP525L) does neither directly nor indirectly interfere with kinesin-1-dependent (anterograde) transport.

This article is well-written and addresses interesting and unclear aspects related to ALS pathology. The methods are sounds. I have only some minor suggestions.

In the discussion oligogenic model of disease in ALS could be briefly addressed reporting associations between FUS variants and TBK1 variants (Lattante et al., Neurobiol Aging 2019;84:239.e9-239.e14). Both FUS and TBK1 have been associated also to frontotemporal dementia (FTD) (Borghero et al., Neurobiol Aging 2016;43:180.e1-5). Remarkably, TBK1 belongs to the category of genes conferring a significantly increased risk but not sufficient to cause disease. According to the oligogenic architecture hypothesized for ALS, I suggest to address this concept emphasizing the role on the clinical phenotype of the potential association between FUS and TBK1 variants or other variants in ALS-related genes.

Author Response

Reviewer 3:

Seifert et al. aimed at investigating whether there is any impact of wildtype FUS and FUS-P525L on anterograde axonal transport, exemplified by in vitro reconstituted microtubule gliding motility assays on surfaces coated with kinesin-1 motors. The Authors used whole-cell lysates obtained from Amyotrophic Lateral Sclerosis (ALS)-patient-specific iPSC-derived spinal motor neurons which endogenously express these FUS variants. They hypothesized that mislocalized pathogenic FUS variants might causally contribute to the observed axonal transport phenotypes of ALS by direct interaction of free FUS with the transport machinery (e.g., directly inhibiting motor proteins or acting as road blocks on microtubules). The results showed that neither recombinant wildtype FUS-GFP nor FUS-P525L-GFP nor cell lysates from motor neurons expressing those FUS variants reduce the microtubule gliding velocity over a wide range of concentrations. Furthermore and in contrast to Tau, wildtype and mutant FUS did not directly bind to microtubules. Hence the Authors concluded that mislocalized cytoplasmic FUS (i.e. FUSP525L) does neither directly nor indirectly interfere with kinesin-1-dependent (anterograde) transport.

This article is well-written and addresses interesting and unclear aspects related to ALS pathology. The methods are sounds. I have only some minor suggestions.

Response: We deeply thank the reviewer for such a positive review!

In the discussion oligogenic model of disease in ALS could be briefly addressed reporting associations between FUS variants and TBK1 variants (Lattante et al., Neurobiol Aging 2019;84:239.e9-239.e14). Both FUS and TBK1 have been associated also to frontotemporal dementia (FTD) (Borghero et al., Neurobiol Aging 2016;43:180.e1-5). Remarkably, TBK1 belongs to the category of genes conferring a significantly increased risk but not sufficient to cause disease. According to the oligogenic architecture hypothesized for ALS, I suggest to address this concept emphasizing the role on the clinical phenotype of the potential association between FUS and TBK1 variants or other variants in ALS-related genes.

Response: Thank you for the suggestion. We have included a paragraph in the discussion where we address this hypothesis.

Reviewer 4 Report

The article presented by Seifert and Colleagues exploits a previously described in-vitro assay aimed at study the kinesin-1-dependent microtubule gliding motility in physiological and pathological conditions. The Authors claimed to obtain two main results: i) to optimise the above in-vitro method in order to tolerate the presence of complex samples such as whole-cell lysates and provides sensitive, highly reproducible and robust readouts of kinesin-1 motility. ii) to demonstrate that the FUS-P525L mutated variants characterising around 5% of ALS familial cases, do not directly bind to Tau or directly interfere with kinesin-1 motility, thus suggesting and indirect effect of mutated FUS in affecting the axonal transport.

The manuscript is well written, results are clearly stated the methods are correctly detailed.

The in-vitro assay used and the results reported related to the direct bind and effects of mutated FUS on axonal transport are very interesting and at least in part unexpected. The only concern it pops up in my mind reading the paper is mainly related to the effective interpretation of the results obtained by the in-vitro method the Authors are using. Although the Author are totally aware of limitations of the methods and they correctly expose this issue, I’m not totally convinced that all the variables  that can affect this in-vitro assay (clearly indicated in the text) can be controlled and do not alter somehow the results. For this reason, as also stated by the Authors, it would be useful for the study to be paralleled by complementary techniques to confirm at least part of the results concerning the pathological section related to ALS FUS variants. Moreover, other ALS mutated proteins (SOD1, TDP43 etc.) are supposed to affect somehow the axonal transport, thus testing their effect on the same in-vitro assay could be another way to show a specific binding with motor proteins or the interference with axonal transport.

Minor comments.

In figure 2, 3 and 4 please indicate, if present, the significance in the bar plot and legend to the figures.

Page 5, line 171 please insert space in “…norFUS-P525Ldirectly…”

The detailed description of the assay method in the first paragraph of the results can be omitted and included in the material and method section only.

Author Response

Reviewer 4:

The article presented by Seifert and Colleagues exploits a previously described in-vitro assay aimed at study the kinesin-1-dependent microtubule gliding motility in physiological and pathological conditions. The Authors claimed to obtain two main results: i) to optimise the above in-vitro method in order to tolerate the presence of complex samples such as whole-cell lysates and provides sensitive, highly reproducible and robust readouts of kinesin-1 motility. ii) to demonstrate that the FUS-P525L mutated variants characterising around 5% of ALS familial cases, do not directly bind to tubulin or directly interfere with kinesin-1 motility, thus suggesting an indirect effect of mutated FUS in affecting the axonal transport.

The manuscript is well written, results are clearly stated the methods are correctly detailed.

Response: We are pleased by this very positive overall statement.

The in-vitro assay used and the results reported related to the direct binding and effects of mutated FUS on axonal transport are very interesting and at least in part unexpected. The only concern it pops up in my mind reading the paper is mainly related to the effective interpretation of the results obtained by the in-vitro method the Authors are using. Although the Author are totally aware of limitations of the methods and they correctly expose this issue, I’m not totally convinced that all the variables  that can affect this in-vitro assay (clearly indicated in the text) can be controlled and do not alter somehow the results. For this reason, as also stated by the Authors, it would be useful for the study to be paralleled by complementary techniques to confirm at least part of the results concerning the pathological section related to ALS FUS variants.

Response: We do very well understand the reviewer’s concern. It is always so difficult to be convinced about negative results, even though these ore often much more valid than „significant differences“. However, as the FUS mutants both in form of the pure FUS protein and cell lysates both showed no difference concerning gliding assay in more than three replicates using different batches of cells or proteins or microtubule preparation, we believe that these results withstand further testing. This is also underpinned by the fact that the positive control (Tau protein) does exactly what was expected. Of note, Tau protein does not only slow gliding assay in case of adding the pure Tau protein, but also does so in cases when Tau is added to the cell lysates, thus additionally showing that there is nothing in the lysates impairing interaction with the microtubules. Together with our general expertise in evaluating gliding assays under various hindering conditions (see e.g. ref 52, DOI: 10.1039/c2lc41099k) we believe that our data justifies the made conclusions.

Moreover, other ALS mutated proteins (SOD1, TDP43 etc.) are supposed to affect somehow the axonal transport, thus testing their effect on the same in-vitro assay could be another way to show a specific binding with motor proteins or the interference with axonal transport.

Response: We do agree with the reviewer that other ALS genes also present axonal phenotypes (DOI: 10.1016/j.nbd.2018.03.010 and 10.1038/sdata.2018.241). However, as shown by us previously, every ALS gene behaves differently concerning axonal phenotypes, with FUS-ALS showing by far the most prominent, strongest and earliest axonal phenotypes in cell culture. This was the reason why we chose FUS-ALS for the current study and even this resulted in negative results. Furthermore, we and others noted that disease mechanisms are significant different between different ALS genes (e.g. doi: 10.3390/ijms21186938), thus we do believe investigating SOD1 and/or TDP43 would be of interest but are clearly separate stories for different manuscripts.

Minor comments.

In figure 2, 3 and 4 please indicate, if present, the significance in the bar plot and legend to the figures.

Page 5, line 171 please insert space in “…norFUS-P525Ldirectly…”

Response: we corrected this.

The detailed description of the assay method in the first paragraph of the results can be omitted and included in the material and method section only.

Response: We very much appreciate the suggestion. However, since the establishment of our modified kinesin-1-dependent microtubule gliding assay presents a partial result of our work, we believe that this description should be included in the results section.